# High-Volume Hemodiafiltration: Expanding the Evidence Beyond Randomized Trials—A Critical Perspective on the 2025 EuDial Consensus

**DOI:** 10.3390/jcm14093174

**Published:** 2025-05-03

**Authors:** Stefano Stuard, Franklin W. Maddux

**Affiliations:** 1Global Medical Office, Fresenius Medical Care, 26020 Palazzo Pignano, Italy; 2Global Medical Office, Fresenius Medical Care, Waltham, MA 02451, USA; frank.maddux@freseniusmedicalcare.com

**Keywords:** high-volume hemodiafiltration, randomized controlled trials, observational studies, mechanistic studies, cardiovascular mortality, inflammation, anemia management, infection-related mortality, intradialytic hypotension, EuDial

## Abstract

The 2025 EuDial Consensus systematically compared hemodiafiltration (HDF) to high-flux hemodialysis (HD), highlighting HDF’s superior removal of middle-molecular-weight uremic toxins, potential survival advantages, and immunomodulatory properties. High-Volume HDF (HVHDF), defined by a substitution volume exceeding 23 L per session, was associated with improved cardiovascular outcomes, reduced infection-related mortality, and decreased systemic inflammation. **Background/Objectives:** Nevertheless, the consensus refrains from endorsing HDF as the standard of care, citing insufficient evidence to prevent sudden cardiac death, reduce intradialytic hypotension, or significantly lower hospitalization rates compared to HD. **Methods:** This review critically evaluates the EuDial Consensus, highlighting its methodological strengths while noting potential limitations stemming from an exclusive reliance on randomized controlled trials (RCTs). The exclusion of real-world evidence (RWE) and mechanistic studies may have led to an underestimation of HDF’s broader clinical benefits, particularly in cardiovascular stability, inflammation control, and anemia management. **Results:** Multiple studies have demonstrated HDF’s capacity to enhance immune function, improve erythropoiesis, and increase the clearance of beta-2 microglobulin (β2M) and other pro-inflammatory toxins. Furthermore, the CONVINCE trial’s economic analysis supports HDF’s cost-effectiveness, especially when considering improved survival and reduced dependency on erythropoiesis-stimulating agents. **Conclusions:** Future research should integrate RWE and mechanistic insights to better define HDF’s therapeutic potential, particularly concerning anemia control, infection mitigation, and hemodynamic stability. While the EuDial Consensus provides valuable clinical guidance, its conclusions should be contextualized within a broader and evolving evidence base. Given its multidimensional benefits, post-dilution HVHDF is increasingly viewed as a preferred renal replacement therapy modality, warranting wider adoption in clinical practice.

## 1. Introduction

Hemodiafiltration (HDF) combines conventional hemodialysis (HD) with convective solute removal through high-volume ultrafiltration and sterile fluid replacement. Post-dilution High-Volume HDF (HVHDF), defined by a substitution volume exceeding 23 L per session, has demonstrated superior removal of middle-molecular-weight uremic toxins and improved patient outcomes compared to high-flux HD.

On 6 February 2025, the European Dialysis (EuDial) Working Group of the European Renal Association (ERA) released a consensus statement assessing the comparative effectiveness of HDF versus high-flux HD in both adult and pediatric populations [1]. This document, grounded in systematic meta-analyses of randomized controlled trials (RCTs) and expert interpretation, evaluated key clinical domains, including all-cause and cardiovascular mortality, cardiovascular events, health-related quality of life (HRQoL), and surrogate biochemical markers. The consensus—formulated into 22 statements—aims to support individualized clinical decision-making without establishing a definitive standard of care.

The EuDial Consensus represents a rigorous, balanced synthesis of the current evidence base. It provides valuable clinical guidance while appropriately delineating the boundaries of the current knowledge. The EuDial Working Group concluded that HDF is associated with improved overall and cardiovascular survival, mainly when delivered at convection volumes exceeding 23 L per session [1]. This observation is consistent with findings from previous prospective interventional studies, including the CONVINCE trial [2], as well as real-world evidence, such as the recent analysis by Zhang et al. [3]. Collectively, these data underscore the clinical importance of achieving an adequate convection volume (>23 L) as a critical determinant of HDF efficacy. The EuDial Working Group consensus also highlighted areas where the current evidence remains inconclusive. No consistent superiority of HDF over high-flux HD was demonstrated in reducing the incidence of sudden cardiac death (SCD), the frequency of intradialytic hypotension (IDH), hospitalization rates, or in improving patient-reported outcomes such as sleep quality and pruritus. These conclusions accurately reflect the heterogeneity and methodological limitations present in the existing literature and underscore the need for continued investigation.

While the Working Group’s conclusions are cautious and evidence-driven, it is important to acknowledge that HDF and HVHDF may exert additional physiological benefits that are not fully captured in RCTs. Specifically, potential advantages in domains such as hemodynamic stability, inflammation modulation, infection-related morbidity, and anemia management remain biologically plausible and are supported by mechanistic data and observational insights. These effects merit further investigation through targeted, adequately powered trials and mechanistic studies.

This review revisits selected aspects, arguing that HVHDF’s advantages—particularly in hemodynamic stability, infection-related outcomes, inflammation control, and anemia management—should not be prematurely dismissed, as potential benefits cannot be excluded at this stage. Continued research is warranted to elucidate the full spectrum of HDF’s clinical impact, particularly as technology and implementation practices continue to evolve.

## 2. Methodological Considerations in the EuDial Consensus

While methodologically rigorous, the EuDial analysis relied exclusively on RCTs. It excluded valuable real-world observational data and mechanistic studies, hindering a more comprehensive assessment of HDF’s effectiveness across diverse clinical settings.

This approach risks downplaying HDF’s potential impact on cardiovascular stability, infection-related mortality, and the reduction in inflammatory markers. Future studies should adopt a hybrid approach, combining RCTs with registry-based data to capture broader patient demographics and real-world effectiveness. Registry-based studies, such as the Dialysis Outcomes and Practice Patterns Study (DOPPS) [4] and EuCliD [5,6], could provide additional insights into long-term outcomes, particularly regarding anemia management and cardiovascular stability. Furthermore, reliance on predefined outcome measures led to the omission of smaller-scale studies that could have explored additional clinical aspects of HDF.

Recently, Canaud et al. emphasized that real-world evidence (RWE) is increasingly acknowledged as a vital component in the assessment of new therapeutic interventions, providing important insights into their effectiveness, safety, and applicability across broader and more representative patient populations [3,7,8,9]. This is particularly relevant in end-stage kidney disease (ESKD), where RCTs—despite being the gold standard for evidence generation—typically include highly selected patient cohorts, adhere to rigid treatment protocols, and involve intensive monitoring. These methodological constraints often limit the external validity of RCT findings and reduce their generalizability to the diverse and complex dialysis population encountered in routine clinical practice [10,11,12,13,14,15,16]. Such concerns are frequently raised by clinicians who question the real-world applicability of RCT results.

## 3. Cardiovascular Stability and Hemodynamic Outcomes

The EuDial Consensus asserts that while HDF may reduce cardiovascular mortality, it does not influence the frequency of IDH episodes [1]. This consensus document considered three trials [17,18,19], of which only one had treatment tolerance as the primary endpoint [18]. In this trial, the risk of asymptomatic hypotension (odds ratio 0.87, 95% CI 0.79–0.95) in patients on online HDF was significantly reduced compared to those on high-flux HD. Concurrently, the risk of symptomatic hypotension episodes was nominally less (odds ratio 0.80, 95% CI 0.60–1.07), but did not achieve statistical significance. In a randomized trial comparing low-flux HD with online pre-dilution HF and/or HDF electing symptomatic hypotension as the primary endpoint, the frequency of sessions with this complication increased for HD (7.1 to 7.9%). In comparison, it decreased for both high-flux HD (9.8 to 8.0%) and HDF (10.6 to 5.2%), and the difference between treatments was highly significant (*p* < 0.001) [20]. Although further studies adopting symptomatic hypotension as a primary endpoint are needed, the available evidence suggests that HDF prevents this dialysis complication.

## 4. Infection-Related Outcomes and Hospitalizations

HDF may enhance immune system function by facilitating the clearance of inflammatory cytokines and improving leukocyte function. In a prospective, open-label, controlled, randomized, crossover pilot study comparing online post-dilution HDF and high-flux HD, pretreatment plasma complement factor D concentrations decreased significantly more with HDF than with high-flux HD (*p* = 0.010) [21]. This factor plays a crucial role in immune defense by regulating the complement system’s alternative pathway, which serves as the first line of defense against infections. Its primary function is to cleave factor B when bound to C3b, forming the C3 convertase (C3bBb), which amplifies complement activation and enhances pathogen clearance [22].

Vaccine studies have demonstrated the immune benefits of HDF compared to other techniques. Patients on HDF exhibited stronger and more sustained antibody responses following vaccinations for influenza and SARS-CoV-2 than those on HD [23,24,25]. The ESHOL trial reported that patients treated with HVHDF had a 55% lower risk of infection-related mortality and a 22% reduction in infection hospitalization rates [17]. The CONVINCE trial identified a 31% lower risk of infection-related death, including cases attributed to COVID-19, among patients receiving HDF [2]. A meta-analysis examining five trials confirmed that patients who achieved high convection volumes experienced significantly lower infection-related mortality [26]. These findings suggest that immunomodulation is a key advantage of HDF that deserves greater recognition.

The EuDial Consensus acknowledges that HDF might reduce infection-related mortality, but suggests that hospitalization rates are comparable between HDF and high-flux HD [1]. However, hospitalization is an intermediate endpoint rather than a direct measure of patient benefit, such as survival, functional status, or quality of life. Treatment may reduce hospital admissions without actually improving patient health or mortality. The hospital admission criteria vary widely across hospitals, regions, and healthcare systems. This variability introduces bias because hospitalization decisions can be influenced by physician discretion, institutional policies, and resource availability rather than the severity of the patient’s condition or the effectiveness of an intervention [27]. Hospitalization does not always correlate directly with disease severity or progression. A patient may be admitted for precautionary reasons rather than significant clinical worsening, which limits its value as a precise measure of treatment efficacy. Conversely, severely ill patients may avoid hospitalization due to palliative care choices or healthcare access issues. The availability of outpatient care, emergency room policies, and insurance coverage can significantly impact hospitalization rates. As a result, hospitalization rates can reflect systemic healthcare practices rather than an actual clinical benefit or harm from a treatment [27]. While analysis of hospitalizations has shown their undisputable impact on health costs and burden of care required, the economic evaluation of the CONVINCE trial demonstrated that the cost–utility ratio of HDF is generally favorable [28].

## 5. Uremic Toxin Removal, Inflammation, and Anemia

According to the EuDial Consensus, nutritional and inflammatory status markers do not differ significantly between HDF and high-flux HD [1].

A comprehensive meta-analysis of the effectiveness of high-flux dialysis and convective dialysis modalities, including HDF and hemofiltration, in removing beta-2-microglobulin (β2M), a middle-molecular-weight solute implicated in dialysis-related amyloidosis and adverse cardiovascular outcomes in end-stage kidney disease (ESKD), was published in 2018 [29]. Given the discrepancies in the literature regarding β2M clearance and the clinical benefits of different dialysis modalities, the authors of this meta-analysis sought to evaluate the determinants of effective β2M removal based on a systematic review of the published studies.

The analysis included 69 studies spanning from 2001 to 2017, incorporating data from 1879 patients and 6771 clearance measurements. Using a random effects meta-analysis and meta-regression model, the authors examined dialysis-related parameters such as membrane composition, modality, blood and dialysate flow rates, and substitution fluid rates [29]. They found that while conventional high flux HD achieved an average β2M clearance of 48.75 mL/min, convective therapies significantly outperformed this, with an average clearance of 87.06 mL/min. HDF provided enhanced clearance, underscoring its potential superiority in removing middle-molecular-weight toxins. Notably, membrane material emerged as a key determinant of β2M clearance. High-flux dialyzers composed of polyarylethersulfone (PAES) exhibited superior β2M clearance in high-flow HD, whereas polysulfone (PS) membranes were associated with better performance in convective therapies such as HDF. The study also highlighted the role of blood flow and substitution fluid rates in optimizing β2M removal. Higher substitution fluid rates in post-dilution HDF resulted in superior clearance, while dialysate flow rates were not found to be a significant factor in enhancing β2M removal.

One intriguing finding was the substantial contribution of adsorption to β2M clearance, particularly when comparing blood-side versus dialysate-side measurements. Adjusted dialysate-side β2M clearances were significantly lower than whole blood clearances, suggesting that membrane adsorption is crucial in trapping β2M beyond diffusive and convective mechanisms. The study found no clear secular trend indicating improved β2M clearance over time, indicating that notwithstanding improved dialysis efficiency, limitations persist in removing β2M effectively [29].

A study based on the randomized controlled CONvective TRAnsport STudy (CONTRAST) investigated the long-term effects of online HDF compared to low-flux HD on systemic inflammation, measured based on high-sensitivity C-reactive protein (CRP) and interleukin-6 (IL-6), in patients with ESKD [30]. The study followed 405 patients for up to three years, analyzing serial measurements of CRP and IL-6, while a broader cohort of 714 patients was assessed for longitudinal changes in serum albumin levels. The results showed that patients undergoing HD exhibited a progressive increase in CRP and IL-6 levels over time, whereas levels remained stable in those treated with HDF. After adjustments for baseline variables, the annual rate of increase in CRP was found to be 20% higher in HD patients, while IL-6 rose by an additional 16% per year relative to those on HDF. The inflammatory advantage of HDF was most pronounced in anuric patients, suggesting that the absence of residual kidney function may amplify the benefits of convective clearance on systemic inflammation. By utilizing a randomized controlled design, CONTRAST provides robust evidence supporting the hypothesis that HDF mitigates systemic inflammation, particularly for CRP and IL-6, without adversely impacting nutritional status, as reflected in the albumin levels. These findings support the broader adoption of HDF as a preferred dialysis modality for reducing inflammation-related complications in the maintenance of dialysis patients.

HDF has shown potential benefits in managing anemia in kidney failure by improving erythropoiesis, reducing inflammation, and removing uremic toxins more efficiently than conventional HD. Studies suggest that HDF enhances the clearance of middle-molecular-weight toxins, some of which are known to inhibit erythroid progenitor cells. For example, HDF with endogenous reinfusion (HFR) has demonstrated a reduction in the suppression of burst-forming unit–erythroid (BFU-E) proliferation [31], which plays a key role in red blood cell production. Furthermore, HDF appears to lower systemic inflammation levels, likely improving erythropoiesis-stimulating agent (ESA) responsiveness and reducing the hepcidin levels that impede iron metabolism [32]. This enhanced toxin removal and reduced inflammatory burden contribute to more effective anemia management. Clinical studies have indicated that HDF can prolong red blood cell lifespan compared to standard HD, leading to more stable hemoglobin levels. One study reported a significant increase in red blood cell survival following a single HDF session [33]. Additionally, patients undergoing HDF have been found to require lower doses of ESAs, suggesting an improvement in iron utilization and erythropoiesis efficiency [34]. Large-scale randomized controlled trials are needed to conclusively determine the magnitude of HDF’s benefit for anemia in ESRD. The overall impact of HDF on anemia management appears to be particularly relevant for patients with inflammation-related ESA resistance, where the reduction in inflammation and improved iron metabolism may play a crucial role in optimizing treatment outcomes.

## 6. Cost-Effectiveness Considerations

Following the publication of the consensus document [1], the CONVINCE investigators reported the cost–utility of high-dose online HDF compared to high-flux HD in patients with kidney failure [28,35]. Given HDF’s established superiority in reducing mortality, the investigators sought to determine whether these clinical benefits justify the increased costs associated with its implementation.

Conducted across 61 centers in eight European countries, the trial randomized 1360 adult patients who had undergone HD for at least three months. Patients were allocated either to high-dose HDF or to continue receiving conventional high-flux HD. The study followed them for at least two years, gathering data on healthcare costs, quality-adjusted life years (QALYs), and broader economic implications [2]. The main results of the trial demonstrated a substantial survival advantage for patients treated with HDF, with a 23% lower risk of all-cause mortality compared to with HD [2]. The cost–utility analysis employed a Markov cohort model to assess outcomes over both a two-year and lifetime horizon, revealing that HDF led to an incremental cost per QALY of €31,898–€37,344 over two years and €27,068–€36,751 over a lifetime [28]. These variations reflect differences in dialysis staff costs across different scenarios. While HDF was associated with greater overall costs due to increased life expectancy and the corresponding need for additional dialysis sessions, sensitivity analyses indicated that at a willingness-to-pay threshold of €50,000 per QALY, the probability of its cost-effectiveness surpassed 90% [28]. When intervention costs in additional life years were excluded, lowering the economic burden of longer survival, the incremental cost-effectiveness ratio dropped significantly to €13,231 per QALY [28]. A key driver of cost differences was the expense of dialysis disposables and increased water and electricity consumption in HDF [28]. However, these costs were partially offset by reduced medication use and potential improvements in patient health-related quality of life [28]. The cost difference between HD and HDF for hospitalizations was relatively small compared with the difference in intervention costs (€374 vs. €603 per year alive), and the risk of recurrent hospitalization for nonfatal causes was not statistically significant [28]; the study acknowledged that more detailed analyses are needed to explore the causes of these admissions. This analysis highlighted variations in cost structures across countries, and the authors recommended that healthcare organizations tailor the findings to local economic conditions [28]. The results emphasize the need to integrate clinical effectiveness with economic viability in treatment decisions, particularly in resource-intensive areas like dialysis care.

Ultimately, the CONVINCE trial presents a compelling case for HDF as a cost-effective therapy for ESKD. While requiring greater initial resource allocation, its demonstrated survival benefits, coupled with its favorable cost-per-QALY ratios, suggest that high-dose HDF represents a valuable investment in patient care.

## 7. Conclusions

The 2025 EuDial Consensus provides a comprehensive and methodologically robust evaluation of HDF, emphasizing its survival and cardiovascular benefits when high convection volumes are consistently achieved. We recognize that the EuDial Consensus statements represent a high tier of evidence and serve as a foundation for clinical decision-making. The consensus appropriately highlights the evidence supporting HVHDF as a modality associated with improved patient outcomes while maintaining a cautious interpretation of findings across domains where the current data remain inconclusive. Notably, the consensus adopts a conservative position in areas with limited randomized controlled trials, such as intradialytic hemodynamic stability, inflammation modulation, and infection-related outcomes. Although the consensus primarily draws on data from randomized trials and meta-analyses, its limited integration of real-world observational studies and mechanistic research may constrain the full appreciation of HDF’s broader physiological and clinical effects. For instance, emerging evidence suggests that HDF may offer additional benefits related to the enhanced clearance of middle molecules, immunomodulatory activity, and improved anemia management—areas that warrant further investigation through targeted research.

The economic analysis from the CONVINCE trial supports HDF’s cost-effectiveness, demonstrating its long-term viability despite its higher upfront resource utilization. In this context, the EuDial Consensus serves as an evidence-informed framework to guide clinical decision-making, while also delineating key knowledge gaps.

Ongoing research should aim to clarify HDF’s role in optimizing inflammation control, infection-related mortality, and other patient-centered outcomes, thereby informing a more nuanced understanding of its place in contemporary dialysis care.

## 8. Perspective

The role of HDF in modern nephrology continues to evolve, with increasing recognition of its potential to address key areas of dialysis-related morbidity. The EuDial Consensus acknowledges its survival benefits, but remains cautious regarding secondary outcomes such as intradialytic hypotension, hospitalization rates, and patient-reported symptoms. This conservatism largely arises from methodological constraints rather than a lack of clinical efficacy. The challenge now lies in refining the evidence base to capture outcomes beyond survival, particularly those influencing patient quality of life and long-term morbidity.

One of the most compelling aspects of HDF is its immunomodulatory effect, which has been shown to reduce systemic inflammatory markers such as CRP and IL-6. Since chronic inflammation is a key driver of cardiovascular disease and anemia in dialysis patients, more in-depth studies should explore how HDF might alter disease progression. The observed enhancement in vaccine response among HDF patients further emphasizes its potential role in improving immune system resilience, a crucial consideration in the face of ongoing challenges like COVID-19 and other infectious diseases.

Additionally, the role of HDF in managing anemia remains a significant area of interest. Traditional erythropoiesis-stimulating agent (ESA) therapies are costly and not universally effective, particularly in patients with inflammation-induced ESA resistance. HDF’s capacity to enhance erythropoiesis by improving the clearance of uremic toxins may reduce dependency on ESAs, potentially leading to substantial cost savings while enhancing patient outcomes. The mechanistic foundations of this effect—especially its influence on hepcidin levels and iron metabolism—require further investigation in large-scale randomized controlled trials.

From an economic standpoint, the findings of the CONVINCE trial represent a paradigm shift. Historically, concerns about HDF’s operational costs have resulted in resistance to its widespread adoption. However, when evaluating cost-effectiveness in terms of quality-adjusted life years (QALYs), HDF appears to provide significant value, particularly in reducing the long-term healthcare expenditures related to infection management, ESA use, and cardiovascular hospitalizations. This encourages policymakers and healthcare organizations to reassess reimbursement models that incentivize HV HDF, ensuring broader patient access to its benefits.

While the EuDial Consensus provides important guidance, future research should continue to explore HDF’s broader benefits, extending beyond mortality reduction. Integrating RWE, refining study designs to capture hemodynamic and immunological effects, and expanding economic analyses will be essential in shaping its role in nephrology. RWE plays a crucial role, not only in guiding clinical decision-making, but also in supporting policy development, particularly in regions or among patient subgroups that have been underrepresented in previous HDF trials.

While post-dilution HVHDF offers significant clinical benefits, it requires blood flow rates that are not feasible in all patients. In contrast, pre-dilution HDF—widely practiced in Japan—achieves high substitution volumes at lower blood flow rates, making it more accessible for patients with vascular access challenges. Kikuchi et al. demonstrated improved survival with pre-dilution HDF compared to HD [36], supporting its consideration as a viable alternative for broader patient populations. Incorporating this modality into clinical discussions may enhance the inclusivity and applicability of HDF.

Given its demonstrated advantages, a shift toward HVHDF as a preferred therapy for patients needing kidney replacement therapy seems increasingly justified.

## Data Availability

No new data were created or analyzed in this study.

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
