# Peer review of "High-Volume Hemodiafiltration: Expanding the Evidence Beyond Randomized Trials—A Critical Perspective on the 2025 EuDial Consensus"

_jcm, 2025, doi:10.3390/jcm14093174_

Round 1

Reviewer 1 Report

Comments and Suggestions for Authors

This is an excellent paper that highlights the limitations of the consensus statement titled “Haemodiafiltration versus high-flux haemodialysis — a Consensus Statement from the EuDial Working Group of the ERA” and offers valuable perspectives for future directions.

However, we kindly request that you revise the manuscript by considering the following points:

High-volume hemodiafiltration (HV-HDF), defined as a substitution volume exceeding 23 liters per session, requires a high blood flow rate, which poses a significant limitation—it cannot be applied to all patients. Therefore, it may not be universally recommended over high-flux hemodialysis (HD) for every patient.

In particular, achieving such high blood flow rates is often challenging in elderly patients and those with diabetes, due to limitations in vascular access.

On the other hand, predilution online HDF, which is commonly performed in Japan, enables high substitution volumes even with lower blood flow rates. This makes it a modality that can be applied to a broader patient population.

We refer to the following study: Kikuchi K, Hamano T, Wada A, Nakai S, Masakane I. Kidney Int. 2019 Apr;95(4):929–938. doi: 10.1016/j.kint.2018.10.036.
Predilution online hemodiafiltration is associated with improved survival compared with hemodialysis.

We recommend that the manuscript also address the potential of predilution HDF and incorporate it into the discussion or perspective.

Author Response

Comments 1: This is an excellent paper that highlights the limitations of the consensus statement titled “Haemodiafiltration versus high-flux haemodialysis — a Consensus Statement from the EuDial Working Group of the ERA” and offers valuable perspectives for future directions. However, we kindly request that you revise the manuscript by considering the following points: High-volume hemodiafiltration (HV-HDF), defined as a substitution volume exceeding 23 liters per session, requires a high blood flow rate, which poses a significant limitation—it cannot be applied to all patients. Therefore, it may not be universally recommended over high-flux hemodialysis (HD) for every patient. In particular, achieving such high blood flow rates is often challenging in elderly patients and those with diabetes, due to limitations in vascular access. On the other hand, predilution online HDF, which is commonly performed in Japan, enables high substitution volumes even with lower blood flow rates. This makes it a modality that can be applied to a broader patient population. We refer to the following study: Kikuchi K, Hamano T, Wada A, Nakai S, Masakane I. Kidney Int. 2019 Apr;95(4):929–938. doi: 10.1016/j.kint.2018.10.036. Predilution online hemodiafiltration is associated with improved survival compared with hemodialysis. We recommend that the manuscript also address the potential of predilution HDF and incorporate it into the discussion or perspective.

Response 1: Thank you for pointing this out. We agree with this comment. We agree that while post-dilution high-volume HDF offers notable advantages, its reliance on high blood flow rates (>330 ml/min x 240 min = 79 L of blood volume processed / session) can present limitations for certain patient groups who may experience vascular access challenges with reduced blood flow rate. In direct response to your suggestions, we have incorporated a discussion of predilution HDF into the manuscript. This addition highlights its widespread adoption in Japan and its valuable ability to achieve substantial convective volumes even at lower blood flow rates. Furthermore, we have included a citation to the significant study by Kikuchi et al. (Kidney Int. 2019;95(4):929–938), which reported improved survival outcomes with predilution HDF compared to conventional HD. This inclusion aims to acknowledge the broader spectrum of HDF modalities and underscore the importance of tailoring treatment strategies to individual patient needs. The following paragraph has been integrated into the Perspective section: “While post-dilution high-volume HDF offers significant clinical benefits, it requires blood flow rates that are not feasible in all patients. In contrast, predilution HDF—widely practiced in Japan—achieves high substitution volumes at lower blood flow rates, making it more accessible for patients with vascular access challenges. Kikuchi et al. demonstrated improved survival with predilution HDF compared to HD, supporting its consideration as a viable alternative for broader patient populations. Incorporating this modality into clinical discussions may enhance the inclusivity and applicability of HDF.”

Reviewer 2 Report

Comments and Suggestions for Authors

The authors have given a thorough review of rhe EuDial Working Group’s consensus guidelines comparing hemodiafiltration (HDF) with high-flux hemodialysis (HD), and they appropriately highlight the need for an integrated synthesis of real-world evidence and randomized controlled trial (RCT) data to inform evidence-based recommendations in the complex clinical context of renal replacement therapy for end-stage kidney disease (ESKD). While this position is broadly supported across medical disciplines, it is important to recognize that clinical practice guidelines—particularly those developed through multidisciplinary consensus, typically hold a higher position on the hierarchy of evidence than narrative reviews. 

Of note, in the section discussing the cost-effectiveness analysis of the CONVINCE trial. The authors state (lines 247–249):

“The cost-utility analysis employed a Markov cohort model to assess outcomes over both a two-year and lifetime horizon, revealing that HDF led to an incremental cost per QALY of €31,898–€37,344 over two years and €27,068–€36,751 over a lifetime.”

Although I am not aware of the most up-to-date cost-effectiveness thresholds used in European settings, traditional benchmarks such as $50,000 per QALY in the U.S. and approximately €50,000 per QALY in Europe are often cited. Based on these values, the long-term cost-effectiveness of HDF appears modest at best, suggesting that meaningful cost savings—particularly through reduction in the price of disposables and accessories—are necessary to improve its value proposition.

The authors rightly note that cost offsets from decreased medication usage should be factored in; however, they omit a critical point—namely, that increased hospitalization rates associated with HDF may contribute to additional healthcare costs, particularly in inpatient settings, that were not accounted for in the cost analysis of the CONVINCE trial. 

Thus, the conclusion of this section appears inconsistent and does not adequately address the significant gaps in our understanding of the long-term economic implications of HDF.

More broadly, the tone of the article implies that healthcare payers should consider perspectives beyond those of a rigorously adjudicated expert panel when evaluating the real-world feasibility of adopting renal replacement modalities that offer modest clinical benefits but are associated with considerable costs, which is very confusing to readers who value the importance of evidence based medicine. Overall, I think the article needs major revisions.

Author Response

Comments 1: The authors have given a thorough review of the EuDial Working Group’s consensus guidelines comparing hemodiafiltration (HDF) with high-flux hemodialysis (HD), and they appropriately highlight the need for an integrated synthesis of real-world evidence and randomized controlled trial (RCT) data to inform evidence-based recommendations in the complex clinical context of renal replacement therapy for end-stage kidney disease (ESKD). While this position is broadly supported across medical disciplines, it is important to recognize that clinical practice guidelines—particularly those developed through multidisciplinary consensus, typically hold a higher position on the hierarchy of evidence than narrative reviews.

Response 1: We thank the reviewer for this insightful comment and for recognizing the scope and intent of our manuscript. We fully agree that the EuDial Consensus Statement, developed through a rigorous multidisciplinary process, holds a high and respected position in the evidence hierarchy. Our objective is not to challenge the authority of such guidelines, but to highlight the potential value of complementing them with RWE and mechanistic data—especially in areas where RCT data remain limited or inconclusive. In lines 57–59, we explicitly recognize the EuDial consensus: “The EuDial consensus represents a rigorous, balanced synthesis of the current evidence base. It provides valuable clinical guidance while appropriately delineating the boundaries of current knowledge”. Our intent is not to challenge this hierarchy but to highlight that (line 74-79): “… HDF and HV-HDF may exert additional physiological benefits that are not fully captured in RCTs. Specifically, potential advantages in domains such as hemodynamic stability, inflammation modulation, infection-related morbidity, and anemia management remain biologically plausible and are supported by mechanistic data and observational insights. These effects merit further investigation through targeted, adequately powered trials and mechanistic studies”. Furthermore, in lines 278–280, we affirm that the consensus “The 2025 EuDial Consensus provides a comprehensive and methodologically robust evaluation of HDF, emphasizing its survival and cardiovascular benefits when high convection volumes are consistently achieved“. In addition, in the lines 282-284 we confirm that “The consensus appropriately highlights the evidence supporting HV-HDF as a modality associated with improved patient outcomes while maintaining a cautious interpretation of findings across domains where current data remain inconclusive.”

Comments 2:  Of note, in the section discussing the cost-effectiveness analysis of the CONVINCE trial. The authors state (lines 247–249): “The cost-utility analysis employed a Markov cohort model to assess outcomes over both a two-year and lifetime horizon, revealing that HDF led to an incremental cost per QALY of €31,898–€37,344 over two years and €27,068–€36,751 over a lifetime.” Although I am not aware of the most up-to-date cost-effectiveness thresholds used in European settings, traditional benchmarks such as $50,000 per QALY in the U.S. and approximately €50,000 per QALY in Europe are often cited. Based on these values, the long-term cost-effectiveness of HDF appears modest at best, suggesting that meaningful cost savings—particularly through reduction in the price of disposables and accessories—are necessary to improve its value proposition.

Response 2: We thank the reviewer for raising this important point. The authors of the EuDial Consensus Document noted the need for a comprehensive cost-utility analysis of the CONVINCE study, one that would consider not only healthcare costs but also those borne by patients, families, and society, including productivity losses: “Limited availability of HDF compared to high-flux HD may compromise equal access to a personalized treatment approach. A cost–utility analysis is necessary to guide the decision-making process between these two modalities” Such an analysis was subsequently published by Schouten et al. (Kidney Int. 2025 Apr;107(4):728–739; doi:10.1016/j.kint.2024.12.018), following the publication date of the EuDial Consensus Statement manuscript. This cost-utility analysis estimated the incremental cost-effectiveness ratio (ICER), expressing the additional cost per quality-adjusted life year (QALY) gained by switching from high-flux HD to high-dose HDF. Specifically, Schouten et al. reported that the probability of HDF being cost-effective exceeded 90% at a willingness-to-pay threshold of €50,000/QALY. Furthermore, when costs associated with additional life years were excluded, the ICER decreased significantly to €13,231/QALY. Importantly, the primary cost drivers were related to increased life expectancy and not to higher costs per dialysis session representing a desirable profile for the therapeutic procedure yielding a beneficial outcome for the patient.

Nephrology Dialysis Transplantation has recently published our comment in response to the EuDial Consensus Document to highlight the above important cost-utility analysis  (Stuard S, Redefining the Value of Hemodiafiltration: A Commentary on the 2025 EuDial Consensus and the CONVINCE Cost-Utility Analysis. Nephrol Dial Transplant, Apr 2025, https://academic.oup.com/ndt/advance-article/doi/10.1093/ndt/gfaf024/8003757, last accessed 24/04/2025). The cost-utility results cited in our manuscript are directly drawn from Schouten et al. (Kidney Int. 2025 Apr;107(4):728–739; doi:10.1016/j.kint.2024.12.018). In accordance with the reviewer’s observation, we have now clarified the attribution in the manuscript of the cost-utility analysis and to provide a more transparent interpretation of the data.

Comments 3: The authors rightly note that cost offsets from decreased medication usage should be factored in; however, they omit a critical point—namely, that increased hospitalization rates associated with HDF may contribute to additional healthcare costs, particularly in inpatient settings, that were not accounted for in the cost analysis of the CONVINCE trial.

Response 3: We thank the reviewer for this important observation. We have now included additional context from Schouten et al., who did address hospitalization in their analysis. As reported in the same study: “We limited our analysis to all-cause hospitalizations. However, the cost difference between HD and HDF for hospitalizations is relatively small compared with the difference in intervention costs (€374 vs. €603 per year alive), and the risk of recurrent hospitalization for nonfatal causes was not statistically significant [28]; The study acknowledged that more detailed analyses are needed to explore the causes of these admissions.” Nonetheless, we agree that further studies are needed to better understand the drivers and financial implications of hospitalization in HDF patients. We have revised the manuscript to reflect this limitation and to recommend this area for future investigation.

Comments 4: Thus, the conclusion of this section appears inconsistent and does not adequately address the significant gaps in our understanding of the long-term economic implications of HDF. More broadly, the tone of the article implies that healthcare payers should consider perspectives beyond those of a rigorously adjudicated expert panel when evaluating the real-world feasibility of adopting renal replacement modalities that offer modest clinical benefits but are associated with considerable costs, which is very confusing to readers who value the importance of evidence based medicine. Overall, I think the article needs major revisions.

Response 4: We sincerely thank the reviewer for this thoughtful feedback and apologize if the tone suggested an unintended devaluation of expert consensus. This is not our intention. In fact, we recognize and fully respect the central role that expert panels and consensus statements play in shaping evidence-based practice. As now emphasized in our revised conclusion, our goal is to  complement—not replace or devalue—the insights provided by the EuDial Consensus. Our revised conclusion now includes the following clarification: “We recognize that EuDial consensus statements represent a high tier of evidence and serve as a foundation for clinical decision-making”. Our review seeks not to replace but to complement these guidelines by highlighting opportunities to incorporate real-world data and mechanistic insights into future recommendations. While the consensus takes a cautious stance in areas where randomized data are limited, it leaves space for emerging evidence—particularly regarding inflammation control, infection-related outcomes, and anemia management. In the conclusion we have stated that the economic analysis from the CONVINCE trial supports HDF's cost-effectiveness, demonstrating its long-term viability despite higher upfront resource utilization.

Round 2

Reviewer 1 Report

Comments and Suggestions for Authors

I would like to thank the authors for their thorough and thoughtful revisions. All the issues raised during the review process have been addressed satisfactorily, and the manuscript has been significantly improved. I recommend it for acceptance.

Reviewer 2 Report

Comments and Suggestions for Authors

The authors have addressed all the issues I raised. I have no more questions/queries/comments.